# Analysis of the Impact of Landscape Patterns on Urban Heat Islands: A Case Study of Chengdu, China

**DOI:** 10.3390/ijerph192013297

**Published:** 2022-10-15

**Authors:** Zishu Sun, Zhigang Li, Jialong Zhong

**Affiliations:** 1Park Urban Research Center, Chengdu University of Technology, Chengdu 610059, China; 2School of Foreign Studies, Chengdu Neusoft University, Chengdu 611844, China; 3College of Management Science, Chengdu University of Technology, Chengdu 610059, China

**Keywords:** landscape configuration, urban heat islands effect, global spatial correlation, Chengdu

## Abstract

The urbanization process, such as population growth and the expansion of roads, railways, residential areas, and industrial areas, causes severe landscape fragmentation and changes in the surface temperature balance, resulting in the heat island effect. This study used Landsat data to study the impact of landscape patterns on urban heat islands (UHIs) and temporal-spatial change characteristics. In addition, spatial correlation analysis was employed to detect the relationships between land surface temperature (LST) and landscape patterns. The results showed that the impervious surfaces landscape area increased significantly, and the Woodland landscape area increased. However, the bare land, cropland, and water body area decreased. The cohesion of cropland and woodland landscape in the suburb decreased, and there was a high degree of fragmentation. The difference between the contributions of the central city and suburbs to the whole region is narrowing, and the expansion of urban heat islands is shifting from the central city to the suburbs. The percentage of landscape index (PLAND) and the patch cohesion index (COHESION) of woodland, water body, and cropland were negatively correlated with LST. Meanwhile, the PLAND and COHESION of impervious surface and bare land were positively correlated with LST, and the splitting index (SPLIT) was the opposite of the PLAND and COHESION. The fragmentation of impervious surfaces and bare land landscapes reduces the UHI effect. Based on these results, countermeasures to mitigate the heat island effect are proposed. These measures will play an essential role in improving urban ecology and the environmental quality of human settlements.

## 1. Introduction

Today, people are increasingly moving to cities, causing the expansion of urban areas around the world [1]. About 54% of the world’s population lives in cities, and the UN expects that figure to rise to 66% by 2050 [2,3]. Population growth has increased the intensity of human activity, and urbanization poses increasingly serious problems. Landscape fragmentation caused by human activity (e.g., the extension of roads, railways, and human settlements) has aggravated the expansion of heat islands and affected local climate conditions. In this context, environmental management decision-makers are paying increasing attention to the sustainable management and restoration of urban and suburban green spaces. China’s 14th Five-Year Plan noted that the ecological green wedge and fan-leaf-shaped layout pattern of the ecological park around the city should be used to create a green heart of the city, forming a multi-level urban public center system, shaping the urban form of the beautiful park, and creating a characteristic corridor space to reduce the heat islands effect.

In 1883, Lake Howard first discovered that the city’s temperature was warmer than that of the surrounding suburbs after comparing the temperature of the city and the suburbs of London [4]. This discovery has attracted widespread attention in academics. In 1958, Manley first proposed the concept of the Urban Heat Island (UHI) [5]. More and more scholars have participated, which has strongly promoted the related research of urban heat islands. At present, the research on UHI is relatively extensive. On the scale, the intensity of surface UHI is estimated based on the global scale and the national scale [6,7]. The environmental sensitivity of surface UHI [8] and the relationship between UHI and outdoor comfort [9] are studied based on local and regional scales. To study the impact of urban green space landscape on seasonal surface temperature based on fine scales such as urban blocks [10] and to study the temporal and spatial changes of the driving factors of UHI [11]. In terms of research methods, the cooling effect of urban vegetation on the UHI is quantified based on the moving crossing method [12], and the characteristics of the UHI are analyzed based on the integration method of mobile measurement and GIS spatial interpolation [13], and the impact of water bodies on the UHI is studied based on the weather research and forecasting model method [14]. Simulation of the UHI effect using machine learning methods [15,16]. Because remote sensing data are relatively easy to obtain, most scholars use optical remote sensing images to evaluate surface UHI. However, a few scholars conduct research by measuring air temperature.

Landscape patterns can change as a result of human activity [17]. The spatial form of the urban landscape determines the urban environment [18], and the increase in the thermal landscape may exacerbate the heat island effect. Common landscape indicators (e.g., patch density, edge density, shape index, aggregation index) have also been used to analyze the effects of green spaces on cooling UHIs [19,20]. Analyzing the shape of the urban landscape through remote-sensing images, combined with landscape measurement, can provide additional accuracy for data representation [21]. The integration of remote sensing and geographic information systems (GIS) is very useful for describing the spatial patterns of urban landscapes. Combining spatial indicators with remote sensing and GIS can facilitate the investigation of the different structural dimensions of urban landscapes, such as location, distribution, size, shape, and arrangement. These are important variables for quantifying urban expansion [22]. Few of these studies spatialized the landscape index to assess the evolution of the spatiotemporal pattern of the landscape.

Regarding the correlation between urban landscape patterns and UHIs, many studies have used land cover data to study the effects of different urban landscapes on UHIs [23,24,25,26,27]. Based on Landsat OLI data, UHIs generated by small and medium-sized cities during different seasons can be quantitatively calculated and their change characteristics analyzed; in this way, the effect of a single land use type on the UHIs of central cities can be studied [28]. One study used the landscape pattern index to analyze trends in the landscape patterns of UHIs in Suqian [29]. Another study summarized the research history and classification of the UHI effect and specifically reviewed the literature on the effect of landscape patterns on UHIs [30]. Others have studied the relationship from the perspective of green space landscapes or green infrastructure. For example, Nastran et al. analyzed the relationship between the size of European UHIs and the range, shape, and distribution of urban green infrastructure using urban households as a unit [31]. Feyisa et al. studied the cooling effect of parks on the thermal environment in large-scale spaces [32]. The cooling effect of parks on their surrounding was positively related to the normalized difference vegetation index (NDVI) and area of park. Others have found that the intensity of UHIs and their correlation with urban green infrastructure are highest in the summer [33,34]. Some studies have found that the intensity of UHIs is most affected by nearby shared green spaces, and considering their spatial distribution is also helpful for analyzing the scope of the influence [35]. Generally, existing studies have shown that agglomerated and continuous large green areas can vigorously promote the reduction of UHIs’ intensity. In addition, these extensive green areas provide more excellent cooling effects than small green areas [36,37]. There were many studies on the correlation coefficient method in previous academic circles. For example, the Pearson correlation coefficient was used to measure the correlation between surface temperature and landscape indicators [38,39]. However, although many scholars have studied the cooling effect of green space, they are rarely combined with spatial correlation analysis technology.

Urban green infrastructure and impervious surfaces are the main components of urban landscapes, and green spaces play a leading role in improving the human living environment. However, rationally configuring green landscapes and impervious surfaces and integrating them with urban planning is a challenging problem. Therefore, this study took the main urban area of Chengdu, China, as the research area and explored the spatial relationship between the main urban landscape pattern and LST using remote sensing data. It also investigated the expansion of the UHIs and considered whether the spatial layout of the landscape is reasonable. Based on the findings, this study makes policy recommendations for improving UHIs and the rational planning of urban landscape patterns. This can help improve the urban living environment.

## 2. Materials and Methods

### 2.1. Research Area and Data Source

The study area of this article is the main urban area of Chengdu, which is an essential part of the Chengdu–Chongqing urban agglomeration. In addition, the study area includes a central city and six suburban areas, as shown in Figure 1. Many scholars at home and abroad use Landsat data to extract urban landscapes, such as Landsat TM, ETM+, OLI, and other data to extract urban green spaces [40,41]. Yang Chaobin et al. used Landsat8 OLI data for land surface temperature inversion and vegetation coverage calculation and analyzed the relationship between green space and the UHI effect [42]. Therefore, Landsat data can meet the requirements of urban landscape pattern extraction and land surface temperature inversion. The Landsat data in this article comes from the geospatial data cloud platform [43]. The data of Landsat 8, when the cloud cover is 0.86%, is selected as the data on 1 May 2017, and the cloud cover of Landsat 7 is 0.16%, and the time was the data on 10 May 2000. The cloud cover of Landsat 5 is 0.32% based on the data of 1 May 1988. The Landsat data obtained from the geospatial data cloud is a level 1 product. The influence of atmospheric water vapor particles has not been eliminated, so FLAASH atmospheric correction is carried out on the image.

### 2.2. Methods

#### 2.2.1. Landscape Information Visualization Methods

According to the “Land Use Classification Standards” and the actual situation of Chengdu’s landscape features, we divide Chengdu’s landscape into the impervious surface, bare land, cropland, water body, and woodland. The impervious surface mainly includes manufactured impervious surfaces and bare rocks. Cropland mainly includes areas covered by vegetation. Finally, bare land mainly includes cropland without vegetation cover and some unused land. The normalized difference index combined with the decision tree method extracts the landscape pattern information in the remote sensing image. The original image and the high-scoring image in Google Earth are used for post-classification processing and evaluation to improve classification accuracy.

The landscape pattern index can accurately reflect the landscape’s structural composition and spatial configuration [44]. Through preliminary research on these indexes provided by FRAGSTATS4.2 software (University of Massachusetts, Amherst, MA, USA) [45], this paper selects the corresponding index from the type and landscape level. In order to show the impact of different types of landscapes on the UHI effect, the splitting index (SPLIT), PLAND, and COHESION at the type level were selected. At the same time, COHESION and SHDI at the landscape level are selected to show spatial variation characteristics of landscape patterns.

#### 2.2.2. Identification of Heat Islands Zone and Analysis of Space-Time Change

UHIs are a wide range of climate effects produced by humans changing the urban atmospheric environment. The land surface temperature (LST) is the most direct manifestation of UHIs. Currently, the study of LST mainly uses remote sensing inversion methods, so many data sources are used for inversions, such as Landsat-TM/OLI data [46,47], NOOA-AVHRR data, and ASTER/MODIS [48,49]. According to the difficulty of data acquisition, this paper chooses the NDVI algorithm, which is currently widely used and can be used for the inversion of LST [50]. In addition, the estimation process described in NASA’s Landsat5 Scientific Data User Manual and NASA’s Landsat8 Data User Manual is used. 

In order to analyze the temporal and spatial changes of the surface heat islands in the study area, the mean-standard deviation method was used to obtain the area range representing different heat levels, and the heat islands area was extracted [51]. Therefore, It is divided into three levels, the high-temperature zone is defined as the urban heat islands area, and the low-temperature zone is the urban green islands area. 

The Heat Islands Distribution Index (HIDI) can describe the contribution of different areas in the study area to the thermal environment of the main urban area [25]. The HIDI has a critical value of 1. When the HIDI is greater than 1, the area’s contribution to the thermal environment is more significant than the average contribution; when the HIDI is less than 1, the area’s contribution to the thermal environment is less than the average contribution.

#### 2.2.3. Spatial Correlation Analysis Method

In order to explore the spatial relationship between the urban landscape pattern and the thermal environment, the Bivariate Moran’s I method was adopted at the genre level in 2017, and the Bivariate Local Moran’s I method at the landscape level was adopted as an example from 1988 to 2017. The correlation between the landscape index and LST and their spatial aggregation changes are studied. The bivariate Moran’s I is statistically significant. This article uses “permutation tests” to evaluate [52,53], which refers to quantifying the observed statistics relative to the “extreme” degree of its distribution under spatial randomness, the principle is to randomly redistribute the observations of one of the variables and recalculate statistics for each such random pattern. The 999 “permutation” is used to test its significance. That is, the number of iterations is 999. The larger the value, the more stable the result is, the less likely it is to cause deviation. The value of spatial correlation significance was set to <0.05.

## 3. Results and Analysis

### 3.1. Analysis of Landscape Area Changes

Figure 2 shows Landscape classification. Calculating the confusion matrix, the classification’s overall accuracy and Kappa coefficient are obtained. For example, the overall accuracy and Kappa coefficient in 1988 were 89.06 and 0.85%, respectively; the overall accuracy and Kappa coefficient in 2000 were 82.26 and 0.79%, respectively; in 2017, the overall accuracy and Kappa coefficient of the year were 93.01 and 0.90%, respectively. 

Figure 3 shows the net change in the landscape area of the study area. From 1988 to 2000, the area of the five landscape types in the study area changed accordingly. The changing area is 67,265.46 ha, accounting for about 23.2% of the total area. The most significant increase in area is impervious surface, an increase of 26,737.56 ha (about 39.75% of the total change area) and an average annual increase of 2228.13 ha. The water body and woodland area increased by 650.34 ha (0.97% of the total change area) and 6244.83 ha (9.28% of the total change area), respectively. The most significant reduction in the area was cropland, which decreased by 25,672.7 ha (38.17% of the total change area), and the average annual reduction was 2139.39 ha. Bare land was reduced by 7960.05 ha (11.83% of the total change area). From 2000 to 2017, the area of various landscapes has undergone tremendous changes. The total area of change has reached 201,216.2 hectares, accounting for 69.41% of the total area of the study area. The area of impervious surfaces still shows a significant increase, increasing by 71,581.59 hectares (accounting for 35.57% of the total change area), and the average annual growth amount reached 4210.68 hectares, nearly doubling from the previous period. The area of ecological land (woodland) also increased considerably by 29,026.53 hectares (accounting for 14.43% of the total change area), and the average annual increase reached 1707.44 hectares; the most significant decrease in the area was cropland, a decrease of 81,812.1 hectares (accounting for 40.66% of the total change area), the average annual decrease reached 4812.47 hectares, and bare land and water body decreased by 17,223.8 hectares, respectively (accounting for 8.56% of the total change area), and 1572.3 hectares (accounting for 0.78% of the total change area).

### 3.2. Analysis of Spatial Distribution and Temporal Changes in the Landscape Pattern Index

Figure 4 shows that the landscape pattern of the study area presented specific typical characteristics. In 1988, the low value of COHESION and the high value of SHDI were mainly concentrated in the connecting areas between urban centers and suburban areas. These areas are mainly expansion areas of human activity. The effect of human activity was relatively strong, and the degree of landscape fragmentation was relatively high. In 2000, the low and high values gradually moved to the outskirts of the urban, and the impervious surface of the urban expanded outward. In 2017, the low values of COHESION and the high value of SHDI were distributed in the suburbs, the landscape connectivity of the cropland and woodland landscapes in the suburbs decreased, and a high degree of fragmentation appeared.

### 3.3. Analysis of Temporal and Spatial Change Trends of Heat Islands 

Figure 5 shows the spatial change pattern of heat island areas in seven regions from 1988 to 2017. In 1988, the UHIs zone was mainly concentrated on the First Ring Road in the Center city (CC). In 2000, the UHIs area of the CC expanded to the first and second ring areas, mainly to the northwest, west, and southwest of the CC. In 2017, the area of UHIs in both the central and suburban areas expanded significantly.

Table 1 shows the UHIs areas, the change in the UHIs areas, and the percentage of change during the three periods of the seven analysis units in the main urban area of Chengdu. From 1988 to 2017, the UHIs area of the seven analysis units gradually increased. Due to rapid urbanization, the average annual increase in the UHIs areas of the seven analysis units from 2000 to 2017 is more than doubled, showing an accelerated growth rate. During period A, the area of the UHIs in the Center city (CC) changed the most, but the percentage of area change was the smallest. The area of the UHIs in the Longquan (LQ) District had the most significant percentage change, while the area of change in the Pidu (PD) District was the smallest. In period B, Xindu (XD) District had the most significant percentage of change in UHIs areas, CC had the slightest percentage change, Shuangliu (SL) District had the most significant area change, and Wenjiang (WJ) District had the slightest change.

In this study, the heat islands distribution index (HIDI) was used to describe the contribution of each area in the main urban area of Chengdu to the thermal environment and the heterogeneity of regional contribution. As shown in Figure 6, there were significant differences in the HIDI in different regions. The HIDI of the three periods in the CC were all greater than 1, indicating that the contribution of this region to the thermal environment was more significant than the average contribution. The HIDI of WJ District, SL District, XD District, and PD District across the three periods were all less than 1, indicating that the contribution of these regions to the thermal environment was less than the average contribution. The HIDI of Qingbaijiang (QBJ) District in 2017 was more significant than 1, and the HIDI in 1988 and 2000 was less than 1. The HIDI of the LQ District in 1988 was less than 1, and the HIDI in 2000 and 2017 was more significant than 1. From 1988 to 2017, with the progress of urbanization, the contribution of the CC to the thermal environment of the main urban area of Chengdu decreased and showed a downward trend, while the contribution of other areas in the suburban area, except WJ District, increased. This also shows that the difference between the contributions of the CC and the suburban area to the main urban area was shrinking, and the expansion of the heat islands area was shifting from the city center to the suburbs.

### 3.4. Analysis of the Impact of Landscape Pattern Changes on the Heat Islands Effect

In order to explore the correlation between different urban landscape patterns (land use and cover types) and LST, this study used the type-level PLAND (i.e., different types to account for the percentage of the unit grid) and SPLIT, and it used the bivariate global Moran’s I index correlation analysis method in GeoDa. Table 2 shows the calculation results for the correlation between the three types of landscape indexes and LST. 

As shown in Table 2, the *p* values of the different landscape types corresponding to the three landscape indexes are all 0.001, passing the 99% significance test (*p* ≤ 0.01). In terms of type, Moran’s I of the PLAND of woodland, water body, and cropland is less than 0, and that of COHESION of woodland, water body, and cropland is also less than 0. Meanwhile, Moran’s I of the PLAND of impervious surfaces and bare land is more significant than 0, and that of COHESION of impervious surfaces and bare land is more significant than 0, which means the decrease in the PLAND and COHESION of woodland, water body, and cropland will increase temperature. Meanwhile, decreasing the PLAND and COHESION of impervious surfaces and bare land will reduce temperature, and the SPLIT will have the opposite effect. It shows that the fragmentation of impervious surfaces and bare land landscapes can reduce the UHI effect. The increase in the patch size of woodland and water body area and increase in COHESION between patches may achieve better results. 

Taking the correlation analysis of PLAND and LST as an example, the correlation between five different landscape types of PLAND and LST is analyzed. As shown in Figure 7, (a) Moran’s I is −0.52, indicating that the water body PLAND is negatively correlated with LST, reflecting that the larger the water body area, the lower the temperature. The water body has a better cooling effect on the urban thermal environment; (b) Moran’s I is −0.48, indicating that the woodland PLAND is negatively correlated with LST, reflecting that the larger the woodland area, the lower the temperature. The woodland has a better cooling effect on the urban thermal environment; (c) Moran’s I is 0.53, indicating that the impervious surfaces PLAND is positively correlated with LST, reflecting that the larger the impervious surfaces area, the higher the temperature. The impervious surfaces has a high warming effect on the urban thermal environment; (d) Moran’s I is 0.21, indicating that the bare land PLAND has a weak positive correlation with LST, and the effect of the bare land landscape on urban heat islands is weak. The bare land landscape has a warming effect on the urban thermal environment; (e) Moran’s I is −0.14, indicating that the cropland landscape has a weak negative correlation with the LST, and the cropland landscape harms the urban heat islands. The contribution of cropland is negative, and the cropland landscape has a cooling effect on the urban thermal environment. Comprehensive analysis of high-temperature landscape is mainly impervious surface and bare land (including cropland without vegetation coverage); low-temperature landscape mainly includes water body, woodland and cropland with high vegetation coverage.

## 4. Discussion

China promulgated the “National Afforestation Planning Outline for 1989–2000,” promoting greening and afforestation throughout the country. Thus, the area of woodland has increased significantly, showing promising results. Meanwhile, the impervious surfaces district has expanded, occupying much cropland. Since 2000, reduced water body, bare land, and cropland have been transformed into impervious surfaces and woodland. The massive increase in impervious surfaces area has also led to the expansion of the UHIs areas and a more apparent thermal environment. The increase in the woodland area is the best embodiment of the achievements in constructing ecological land (woodland) in the study area.

The main urban area of Chengdu is the core area of Chengdu’s urban agglomeration, with a relatively dense population, rapid economic development, an urbanization rate greater than 70%, and a resident population of approximately 16 million. With the acceleration of urbanization and industrialization, many ecological lands such as woodland and cropland in the study area are used for the construction of infrastructure, and other significant projects, resulting in profound loss of natural resources and apparent expansion of urban heat islands.

The research on the correlation between different types of landscapes and surface temperature shows that forest land and water area have a high negative correlation with temperature. These cold landscapes have a slowing effect on the urban surface heat island. This result is consistent with previous studies on the regional scale [14,34,54]. Therefore, the connectivity of these landscapes and their landscape percentages (PLAND) should be increased. For example, in the woodland landscape of urban boulevards, trees with better canopy cover can be selected to improve the shading effect. However, different trees have different shading capabilities. One crucial criterion is that when considering shading effects, the effect of vegetation on urban airflow must also be considered. Therefore, choosing trees with a higher canopy and better canopy coverage can improve the shading effect and ensure airflow. Further, building connected ditches on both sides of impervious roads or the middle green belt to introduce flowing water can increase urban air humidity and improve the microclimate. The analysis of large and small ditches in the urban area of Chengdu indicated that the water sources in some areas had poor fluidity and appeared to be dry. Treating these areas and introducing flowing water can help alleviate UHIs. The layout of woodland and water body landscapes should be effectively configured and combined, further mitigating the UHI effect. The PLAND of the impervious surfaces is positively correlated with the LST, reflecting that when the impervious surface area in the region is larger, there will be a higher temperature. The impervious surfaces contribute more to the urban thermal environment. This result is consistent with the results of previous studies [24,55]. Therefore, the SPLIT of a thermal landscape should be increased. The low value of COHESION is mainly distributed in suburban. These low-value gathering areas are mainly distributed in cold landscapes such as waters and woodlands, and the fragmentation is severe. Therefore, it is necessary to protect the existing cooling landscapes in the suburbs, such as large forest farms, wetlands, and parks, and to treat bare land landscapes are treated with tree-planting or grass-planting to reduce the area of bare land and control the expansion of the impervious surfaces in the built-up area. Reasonable allocation of woodland and water body landscapes, an effective combination of the two layouts, improves the urban heat island effect.

## 5. Conclusions 

From the perspective of urban landscape planning, this study used Landsat data to study the impact of landscape patterns on urban heat islands and the characteristics of temporal-spatial changes. In addition, spatial correlation analysis was employed to detect the relationships between land surface temperature (LST) and landscape patterns. The main conclusions are as follows.

From 1988 to 2017, the impervious surfaces landscape area of the main urban area in Chengdu had increased significantly, the woodland landscape area had increased, and the bare land, cropland, and water body areas decreased. The reduced cropland, bare land, and water body were mainly transformed into impervious surfaces and woodland. As a result, the city’s impervious surfaces has expanded outward, the COHESION of cropland and woodland landscapes in the suburbs was reduced, and there was a high degree of fragmentation.

From 1988 to 2017, the UHIs area of the seven analysis units gradually increased. However, due to rapid urbanization, the average annual increase in the UHIs areas of the seven analysis units from 2000 to 2017 is more than doubled, showing an accelerated growth rate. The central city area’s contribution to the thermal environment of the main urban area showed a downward trend, and the contribution of other areas, except the WJ district in the suburban area, showed an upward trend. This also reflects the shrinking trend in the difference between the urban center and the suburban area’s contribution to the entire area, and the expansion of the UHIs area has shifted from the urban center to the suburbs.

The type-level global Moran’s I spatial correlation analysis showed that the PLAND of woodland, water body, and cropland were negatively correlated with COHESION. Such weakening will lead to an increase in temperature. Meanwhile, the PLAND and COHESION of impervious surfaces and bare land showed a positive correlation, and its weakening will reduce the temperature, while the SPLIT will have the opposite result. The degree of correlation between different landscape types of indices and temperature also differed significantly. High-temperature landscapes are mainly impervious surfaces and bare land, while low-temperature landscapes mainly include water bodies, woodland, and cropland with high vegetation coverage.

## Figures and Tables

**Figure 1 ijerph-19-13297-f001:**
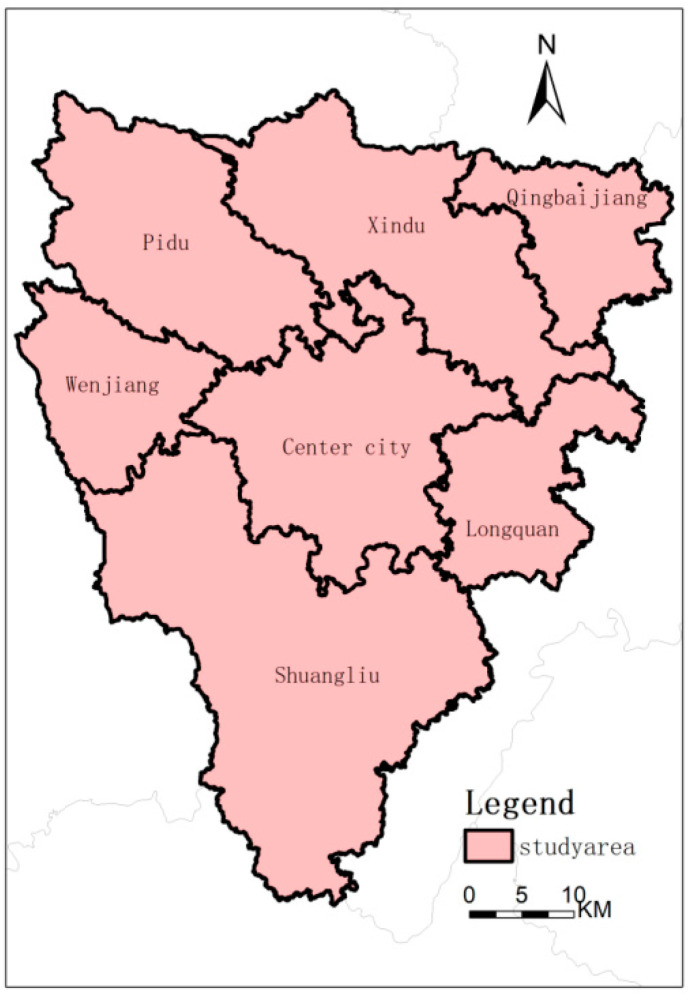
It is an administrative boundary map of the main urban area, including seven secondary administrative regions: Center city, Pidu, Xindu, Wenjiang, Shuangliu, Longquan, Xindu, and Qingbaijiang.

**Figure 2 ijerph-19-13297-f002:**
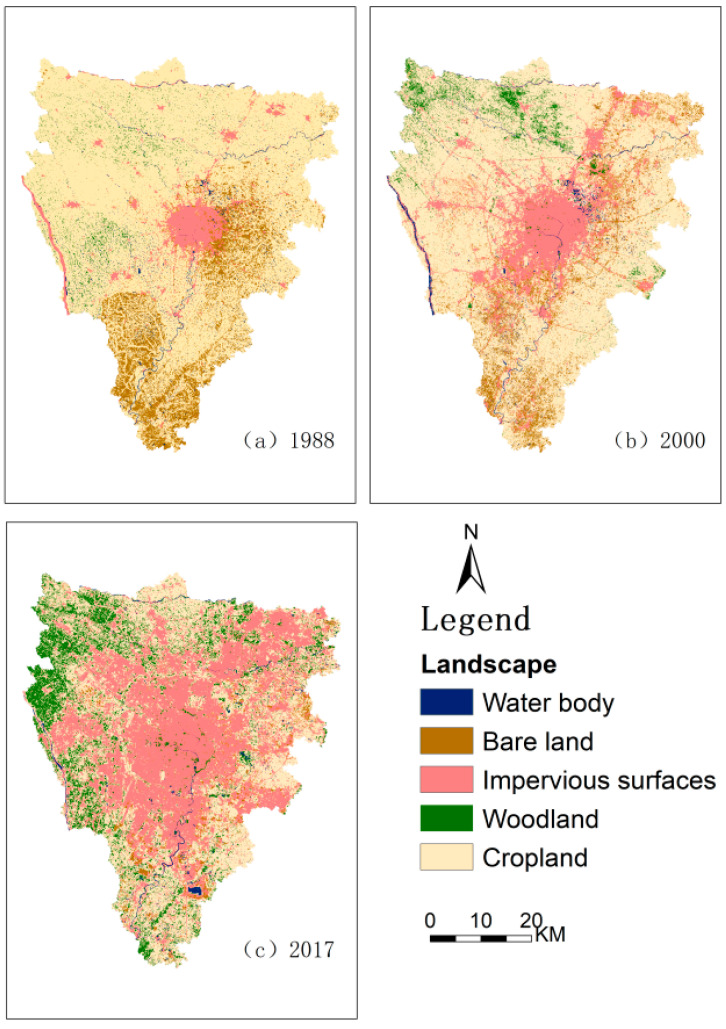
**(a–c)** **are the landscape classification maps in 1988, 2000, and 2017 respectively.**

**Figure 3 ijerph-19-13297-f003:**
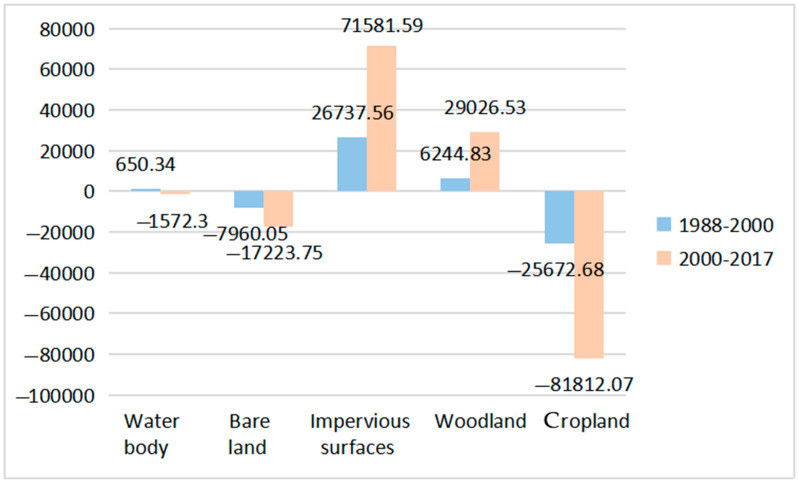
The net change in landscape area in the study area.

**Figure 4 ijerph-19-13297-f004:**
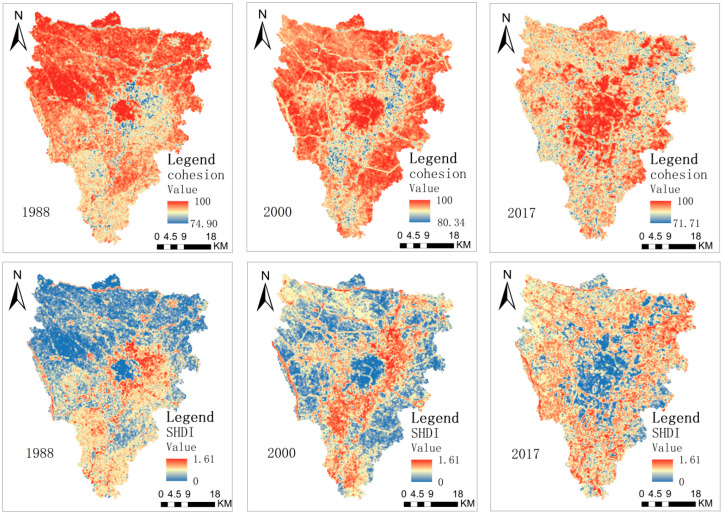
The spatial distribution of landscape index in 1988, 2000, and 2017.

**Figure 5 ijerph-19-13297-f005:**
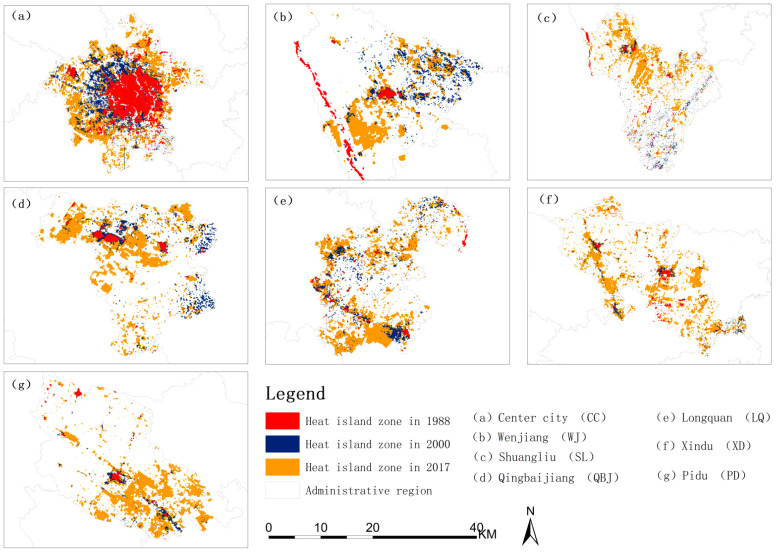
In 1988, 2000, and 2017 urban heat islands area expansion map. (**a**) is the heat island zone of Center city in 1988, 2000, and 2017; (**b**) is the heat island zone of Wenjiang in 1988, 2000, and 2017; (**c**) is the heat island zone of Shuangliu in 1988, 2000, and 2017; (**d**) is the heat island zone of Qingbaijiang in 1988, 2000, and 2017; (**e**) is the heat island zone of Longquan in 1988, 2000, and 2017; (**f**) is the heat island zone of Xindu in 1988, 2000, and 2017; (**g**) is the heat island zone of Pidu in 1988, 2000, and 2017.

**Figure 6 ijerph-19-13297-f006:**
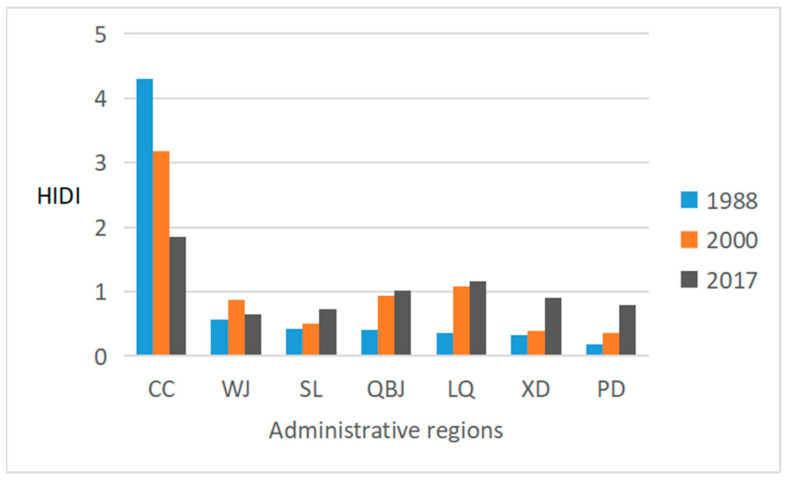
The HIDI of each analysis unit in the main urban area of Chengdu in 1988, 2000, and 2017.

**Figure 7 ijerph-19-13297-f007:**
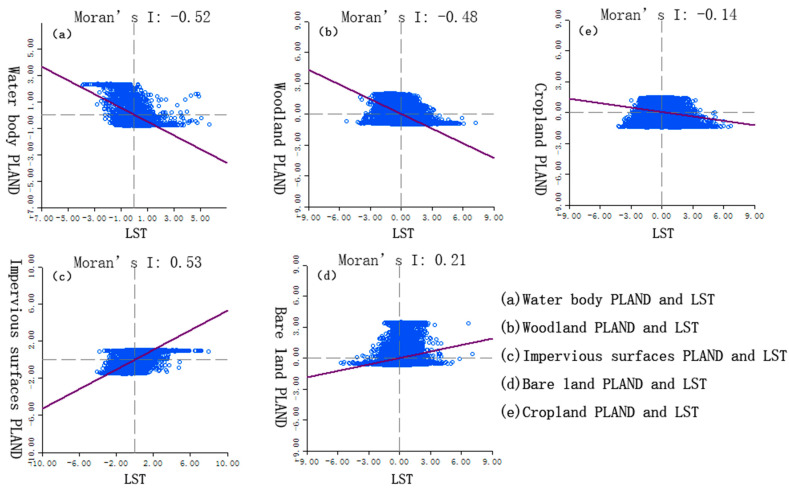
The correlation between five different types of landscape and LST. (**a**) is the correlation between water body PLAND and LST; (**b**) is the correlation between woodland PLAND and LST; (**c**) is the correlation between impervious surfaces PLAND and LST; (**d**) is the correlation between bare land PLAND and LST; (**e**) is the correlation between cropland PLAND and LST.

**Table 1 ijerph-19-13297-t001:** Changes in the area of the heat islands area in the main urban area of Chengdu.

Time	1988	2000	2017	A (1988–2000) Change		B (2000–2017) Change	
Zone	Heat islands area (ha)			Area (ha)	%	Area (ha)	%
CC	8804.25	11,013.12	22,781.61	2208.87	25.09	11,768.49	106.86
QBJ	384.21	1472.04	5635.71	1087.83	283.13	4163.67	282.85
XD	699.57	1460.16	11,906.8	760.59	108.72	10,446.64	715.44
PD	345.78	1099.8	8742.51	754.02	218.06	7642.71	694.92
WJ	566.91	1473.84	3861.09	906.93	159.98	2387.25	161.97
SL	1565.91	3148.92	16,290.90	1583.01	101.09	13,141.98	417.35
LQ	376.83	1879.92	7203.96	1503.09	398.88	5324.04	283.21
Total	12,743.46	21,547.80	76,422.58	8804.34	69.09	54,874.78	254.67

**Table 2 ijerph-19-13297-t002:** The correlation between the landscape index and LST.

Landscape Index	Landscape Type	Moran’s I	*p*	Z
COHESION	Impervious surface	0.47	0.001	259.12
	Barren	0.17	0.001	51.93
	Water body	−0.42	0.001	−47.93
	Woodland	−0.42	0.001	−174.99
	Cropland	−0.14	0.001	−89.64
SPLIT	Impervious surface	−0.36	0.001	−197.31
	Barren	−0.15	0.001	−48.38
	Water body	0.34	0.001	40.21
	Woodland	0.37	0.001	155.79
	Cropland	0.13	0.001	80.54
PLAND	Impervious surface	0.53	0.001	311.63
	Barren	0.21	0.001	76.58
	Water body	−0.48	0.001	−64.99
	Woodland	−0.52	0.001	−232.90
	Cropland	−0.14	0.001	−111.33

## Data Availability

The data that support the findings of this study are available from the corresponding author upon reasonable request.

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
