# Peer review of "Analysis of the Impact of Landscape Patterns on Urban Heat Islands: A Case Study of Chengdu, China"

_ijerph, 2022, doi:10.3390/ijerph192013297_

Round 1

Reviewer 1 Report

First of all, the topic of the paper does not match its content. The main contribution of the paper is to use LANDSAT data to analyze the temporal and spatial change of land surfaces and UHI effects in a Chinese city. Not so much on "countermeasures for reducing the UHI effects".

The paper is complete and generally correct technically. However, some basic concepts, equations, and techniques to process LANDSAT data should not be presented in length. They are existing knowledge.

Figure 7 is weird. What is the blue straight line in the graph?

The literature part should be strenghthened. Some important works, especially reviews, related to the paper are missing. Go to Building and Environment and other renowned journals. Search for those reviews on urban thermal environment, urban heat island, etc.

English language should be improved.

Author Response

Dear reviewer,

Thank you very much for your valuable time and comments. We were pleased to have an opportunity to revise our paper entitled “Analysis of the Impact of Landscape Patterns on Urban Heat Islands: A Case Study of Chengdu, China”. As instructed, we have carefully considered the reviews and replied to each comment in a point-by-point fashion.

Point 1: First of all, the topic of the paper does not match its content. The main contribution of the paper is to use LANDSAT data to analyze the temporal and spatial change of land surfaces and UHI effects in a Chinese city. Not so much on "countermeasures for reducing the UHI effects".

Response 1: Thanks so much for the suggestion. This study used Landsat data to study the impact of landscape patterns on urban heat islands (UHIs) and temporal-spatial changes characteristics. Therefore, we have revised the title of the article to "Analysis of the Impact of Landscape Patterns on Urban Heat Islands: A Case Study of Chengdu, China" to match the theme with the content better.

Point 2: The paper is complete and generally correct technically. However, some basic concepts, equations, and techniques to process LANDSAT data should not be presented in length. They are existing knowledge.

Response 2: Thanks so much for the suggestion. To avoid a lengthy introduction, we rewritten the structure about “2.2. Methods”. Some basic concepts, equations, and techniques for processing LANDSAT data are included in the“Appendix A” file for additional submission.
2.2. Methods

2.2.1. Landscape information visualization methods

According to the ‘Land Use Classification Standards’ and the actual situation of Chengdu's landscape features, we divide Chengdu's landscape into the impervious surface, bare land, cropland, water body, and woodland. The impervious surface mainly includes manufactured impervious surfaces and bare rocks. Cropland mainly includes areas covered by vegetation. Finally, bare land mainly includes cropland without vegetation cover and some unused land. The normalized difference index combined with the decision tree method extracts the landscape pattern information in the remote sensing image. The original image and the high-scoring image in Google Earth are used for post-classification processing and evaluation to improve classification accuracy.

The landscape pattern index can accurately reflect the landscape's structural composition and spatial configuration [43]. Through preliminary research on these indexes provided by FRAGSTATS4.2 software [44], this paper selects the corresponding index from the type and landscape level. In order to show the impact of different types of landscapes on the UHI effect, the splitting index (SPLIT), PLAND, and COHESION at the type level were selected. At the same time, COHESION and SHDI at the landscape level are selected to show spatial variation characteristics of landscape patterns.

2.2.2 Identification of heat islands zone and analysis of space-time change

UHIs are a wide range of climate effects produced by humans changing the urban atmospheric environment. The land surface temperature (LST) is the most direct manifestation of UHIs. Currently, the study of LST mainly uses remote sensing inversion methods, so many data sources are used for inversions, such as Landsat-TM/OLI data [45,46], NOOA-AVHRR data, and ASTER/MODIS [47,48]. According to the difficulty of data acquisition, this paper chooses the NDVI algorithm, which is currently widely used and can be used for the inversion of LST [49]. In addition the estimation process described in NASA’s Landsat5 Scientific Data User Manual and NASA’s Landsat8 Data User Manual is used.

In order to analyze the temporal and spatial changes of the surface heat islands in the study area, the mean-standard deviation method was used to obtain the area range representing different heat levels, and the heat islands area was extracted [50]. Therefore, It is divided into three levels, and the high-temperature zone is defined as the urban heat islands area, and the low-temperature zone is the urban green islands area.

The Heat islands Distribution Index (HIDI) can describe the contribution of different areas in the study area to the thermal environment of the main urban area [24]. The HIDI has a critical value of 1. When the HIDI is greater than 1, the area’s contribution to the thermal environment is more significant than the average contribution; when the HIDI is less than 1, the area’s contribution to the thermal environment is less than the average contribution.

2.2.3. Spatial correlation analysis method

  In order to explore the spatial relationship between the urban landscape pattern and the thermal environment, the Bivariate Moran's I method was adopted at the genre level in 2017, and the Bivariate Local Moran's I method at the landscape level was adopted as an example from 1988 to 2017. The correlation between the landscape index and LST and their spatial aggregation changes are studied. The bivariate Moran's I is statistically significant. This article uses "permutation tests" to evaluate [51,52], which refers to quantifying the observed statistics Relative to the "extreme" degree of its distribution under spatial randomness, the principle is to randomly redistribute the observations of one of the variables, and recalculate statistics for each such random pattern. The 999 "permutation" is used to test its significance. That is, the number of iterations is 999. The larger the value, the more stable the result is, the less likely it is to cause deviation. Set the value of spatial correlation significance to <0.05.

Point 3:Figure 7 is weird. What is the blue straight line in the graph?

Response 3: The blue straight line in Figure 7 is the regression line fitted by the Moran scatter plot, and the slope of this blue line is the Moran’s I index.

Point 4:The literature part should be strenghthened. Some important works, especially reviews, related to the paper are missing. Go to Building and Environment and other renowned journals. Search for those reviews on urban thermal environment, urban heat island, etc.

Response 4: Thanks so much for the suggestion. We have added some reviews on urban heat islands in lines 46-65 in Introduction and discussed them separately regarding scale, methods, and data selection.

In 1883, Lake Howard first discovered that the city’s temperature was warmer than that of the surrounding suburbs after comparing the temperature of the city and the suburbs of London [3]. This discovery has attracted widespread attention in academics. In 1958, Manley first proposed the concept of the Urban Heat Islands (UHIs) [4]. More and more scholars have participated, which has strongly promoted the related research of urban heat islands. At present, the research on UHIs is relatively extensive. On the scale, the intensity of surface UHIs is estimated based on the global scale and the national scale [5,6]. The environmental sensitivity of surface UHIs [7] and the relationship between UHIs and outdoor comfort [8] are studied based on local and regional scales. To study the impact of urban green space landscape on seasonal surface temperature based on fine scales such as urban blocks [9] and to study the temporal and spatial changes of the driving factors of UHIs [10]. In terms of research methods, the cooling effect of urban vegetation on the UHIs is quantified based on the moving crossing method [11], the characteristics of the UHIs are analyzed based on the integration method of mobile measurement and GIS spatial interpolation [12], and the impact of water bodies on the UHIs is studied based on the weather research and forecasting model method [13]. Simulation of the UHI effect using machine learning methods [14,15]. Because remote sensing data is relatively easy to obtain, most scholars use optical remote sensing images to evaluate surface UHIs. However, a few scholars conduct research by measuring the air temperature.

Point 5:English language should be improved.  

Response 5: Thanks so much for the suggestion. We have sent the article to a language editing agency for language editing.

Kind regards,

The Authors

Reviewer 2 Report

In this manuscript, the authors attempted to monitor the changes of landscape and propose measures to reduce urban heat islands by using Landsat images. Unfortunately, the manuscript suffers from very poor grammar,  limited physical explanations, and an overall lack of discussion as to how this study builds upon the large body of studies around the world. The content of the manuscript is not, for the most part, novel other than to say that it covers a previously uncovered geographic region, and therefore contributes little to the spatial-temporal analysis. At this time, I can't find real highlights in present form. The following issues should be clarified!

Major comments:

1 Scientific relevance: In my opinion, the major issue is that the authors do not provide specific scientific questions that this study would like to address. I would suggest the author further outline the context or question for this study so that the scientific relevance of this study is much clearer.

2 Introduction: The authors should assess and summarize the current references appropriately in the introduction part.

3 Methods:  The normalized index was not the best choice to obtain the landscape information.

4 Results:

The information in Fig.2 was not accurate. How to explain there was little woodland in year 1980? How to understand Fig.5?

5 Discussion:Very little discussion is offered contrasting existent results with similar studies.

All the figures are not clear enough!

It is strongly recommended to proofread the paper thoroughly again.

Author Response

Dear reviewer,

Thank you very much for your valuable time and comments. We were pleased to have an opportunity to revise our paper entitled “Analysis of the Impact of Landscape Patterns on Urban Heat Islands: A Case Study of Chengdu, China”. As instructed, we have carefully considered the reviews and replied to each comment in a point-by-point fashion.

Point 1: Scientific relevance: In my opinion, the major issue is that the authors do not provide specific scientific questions that this study would like to address. I would suggest the author further outline the context or question for this study so that the scientific relevance of this study is much clearer.
Response 1: Currently, many scholars study the cooling effect of green space, mainly using the Pearson correlation coefficient to measure, but it is less combined with the spatial correlation analysis technology. This study used Landsat data to study the impact of landscape patterns on urban heat islands (UHIs) and temporal-spatial changes characteristics. In addition, spatial correlation analysis was employed to detect the relationships between land surface temperature (LST) and landscape patterns.

Point 2: Introduction: The authors should assess and summarize the current references appropriately in the introduction part.

Response 2: Thanks so much for the suggestion. We have added some reviews on urban heat islands in the Introduction and discussed them separately regarding scale, methods, and data selection. Current references are appropriately assessed and summarized.

1. Introduction

Today, people are increasingly moving to cities, causing the expansion of urban areas around the world[1]. About 54% of the world’s population lives in cities, and the UN expects that figure to rise to 66% by 2050 [2]. Population growth has increased the intensity of human activity, and urbanization poses increasingly serious problems. Landscape fragmentation caused by human activity (e.g., the extension of roads, railways, and human settlements) has aggravated the expansion of heat islands and affected local climate conditions. In this context, environmental management decision-makers are paying increasing attention to the sustainable management and restoration of urban and suburban green spaces. China’s 14th Five-Year Plan noted that the ecological green wedge and fan-leaf-shaped layout pattern of the ecological park around the city should be used to create a green heart of the city, forming a multi-level urban public center system, shaping the urban form of the beautiful park, and creating a characteristic corridor space to reduce the heat-islands effect.

In 1883, Lake Howard first discovered that the city’s temperature was warmer than that of the surrounding suburbs after comparing the temperature of the city and the suburbs of London [3]. This discovery has attracted widespread attention in academics. In 1958, Manley first proposed the concept of the Urban Heat Island (UHI) [4]. More and more scholars have participated, which has strongly promoted the related research of urban heat islands. At present, the research on UHI is relatively extensive. On the scale, the intensity of surface UHI is estimated based on the global scale and the national scale [5,6]. The environmental sensitivity of surface UHI [7] and the relationship between UHI and outdoor comfort [8] are studied based on local and regional scales. To study the impact of urban green space landscape on seasonal surface temperature based on fine scales such as urban blocks [9] and to study the temporal and spatial changes of the driving factors of UHI [10]. In terms of research methods, the cooling effect of urban vegetation on the UHI is quantified based on the moving crossing method [11], the characteristics of the UHI are analyzed based on the integration method of mobile measurement and GIS spatial interpolation [12], and the impact of water bodies on the UHI is studied based on the weather research and forecasting model method [13]. Simulation of the UHI effect using machine learning methods [14,15]. Because remote sensing data is relatively easy to obtain, most scholars use optical remote sensing images to evaluate surface UHI. However, a few scholars conduct research by measuring the air temperature.

Landscape patterns can change as a result of human activity [16]. The spatial form of the urban landscape determines the urban environment [17], and the increase in the thermal landscape may exacerbate the heat island effect. Common landscape indicators (e.g., patch density, edge density, shape index, aggregation index) have also been used to analyze the effects of green spaces on cooling UHIs [18,19]. Analyzing the shape of the urban landscape through remote-sensing images, combined with landscape measurement, can provide additional accuracy for data representation [20]. The integration of remote sensing and geographic information systems (GIS) is very useful for describing the spatial patterns of urban landscapes. Combining spatial indicators with remote sensing and GIS can facilitate investigating the different structural dimensions of urban landscapes, such as location, distribution, size, shape, and arrangement. These are important variables for quantifying urban expansion [21]. A few of these studies spatialized the landscape index to assess the evolution of the spatiotemporal pattern of the landscape.

Regarding the correlation between urban landscape patterns and UHIs, many studies have used land-cover data to study the effects of different urban landscapes on UHIs [22,23,24,25,26]. Based on Landsat–OLI data, UHIs generated by small and medium-sized cities during different seasons can be quantitatively calculated and their change characteristics analyzed; in this way, the effect of a single land-use type on the UHIs of central cities can be studied [27]. One study used the landscape pattern index to analyze trends in the landscape patterns of UHIs in Suqian [28]. Another study summarized the research history and classification of the UHI effect and specifically reviewed the literature on the effect of landscape patterns on UHIs [29]. Others have studied the relationship from the perspective of green space landscapes or green infrastructure. For example, Nastran et al. analyzed the relationship between the size of European UHIs and the range, shape, and distribution of urban green infrastructure using urban households as a unit [30]. Feyisa et al. studied the cooling effect of parks on the thermal environment in large-scale spaces [31]. The cooling effect of parks on the surrounding environment was positively correlated with normalized difference vegetation index (NDVI) and park area. Others have found that the intensity of UHIs and their correlation with urban green infrastructure are highest in the summer [32,33]. Some studies have found that the intensity of UHIs is most affected by nearby shared green spaces, and considering their spatial distribution is also helpful for analyzing the scope of the influence [34]. Generally, existing studies have shown that agglomerated and continuous large green areas can vigorously promote the reduction of UHIs intensity. In addition, these extensive green areas provide more excellent cooling effects than small green areas [35,36]. There were many studies on the correlation coefficient method in previous academic circles. For example, the Pearson correlation coefficient was used to measure the correlation between surface temperature and landscape indicators [37,38]. However, although many scholars have studied the cooling effect of green space, they are rarely combined with spatial correlation analysis technology.

Urban green infrastructure and impervious surfaces are the main components of urban landscapes, and green spaces play a leading role in improving the human living environment. However, rationally configure green landscapes and impervious surfaces and integrate them with urban planning is a challenging problem. Therefore, this study took the main urban area of Chengdu, China, as the research area and explored the spatial relationship between the main urban landscape pattern and LST using remote-sensing data. It also investigated the expansion of the UHIs and considered whether the spatial layout of the landscape is reasonable. Based on the findings, this study makes policy recommendations for improving UHIs and the rational planning of urban landscape patterns. This can help improve the urban living environment.

Point 3: Methods:  The normalized index was not the best choice to obtain the landscape information.

Response 3: The normalized difference index combined with the decision tree method extracts the landscape pattern information in the remote sensing image. The original image and the high-scoring image in Google Earth are used for post-classification processing and evaluation to improve classification accuracy.

Point 4: Results:The information in Fig.2 was not accurate. How to explain there was little woodland in year 1980? How to understand Fig.5?

Response 4: In Fig.2, in 1988, the surrounding area of Chengdu was mainly cropland. However, since 1999, Chengdu has implemented the policy of returning cropland to woodland, so the woodland area has increased. We modified Fig.5 to analyze the seven regions' Spatio-temporal changes of heat islands. From 2000 to 2017, the heat island expanded significantly, which was caused by rapid urbanization.

Point 5: Discussion:Very little discussion is offered contrasting existent results with similar studies.

Response 5: Thanks so much for the suggestion. In the Discussion section, we have made improvements to compare our findings with similar studies and propose deploying mitigation measures in lines 346-376.

4. Discussion

China promulgated the “National Afforestation Planning Outline for 1989–2000,” promoting greening and afforestation throughout the country. Thus, the area of woodland has increased significantly, showing promising results. Meanwhile, the impervious surfaces district has expanded, occupying much cropland. Since 2000, reduced water body, bare land, and cropland have been transformed into impervious surfaces and woodland. The massive increase in impervious surfaces area has also led to the expansion of the UHIs areas and a more apparent thermal environment. The increase in the woodland area is the best embodiment of the achievements in constructing ecological land (woodland) in the study area.

The main urban area of Chengdu is the core area of Chengdu’s urban agglomeration, with a relatively dense population, rapid economic development, an urbanization rate greater than 70%, and a resident population of approximately 16 million. With the acceleration of urbanization and industrialization, many ecological lands such as woodland and cropland in the study area are used for the construction of infrastructure, and other significant projects, resulting in profound loss of natural resources and apparent expansion of urban heat islands.

The research on the correlation between different types of landscapes and surface temperature shows that forest land and water area have a high negative correlation with temperature. These cold landscapes have a slowing effect on the urban surface heat island. This result is consistent with previous studies on the regional scale [13,53,54]. Therefore, the connectivity of these landscapes and their landscape percentages (PLAND) should be increased. For example, in the woodland landscape of urban boulevards, trees with better canopy cover can be selected to improve the shading effect. However, different trees have different shading capabilities. One crucial criterion is that when considering shading effects, the effect of vegetation on urban airflow must also be considered. Therefore, choosing trees with a higher canopy and better canopy coverage can improve the shading effect and ensure airflow. Further, building connected ditches on both sides of impervious roads or the middle green belt to introduce flowing water can increase urban air humidity and improve the microclimate. The analysis of large and small ditches in the urban area of Chengdu indicated that the water sources in some areas had poor fluidity and appeared to be dry. Treating these areas and introducing flowing water can help alleviate UHIs. The layout of woodland and water body landscapes should be effectively configured and combined, further mitigating the UHI effect. The PLAND of the impervious surfaces is positively correlated with the LST, reflecting that when the impervious surfaces area in the region is larger, there will be a higher temperature. The impervious surfaces contribute more to the urban thermal environment. This result is consistent with the results of previous studies[55,56]. Therefore, the SPLIT of a thermal landscape should be increased. The low value of COHESION is mainly distributed in suburban. These low-value gathering areas are mainly distributed in cold landscapes such as waters and woodlands, and the fragmentation is severe. Therefore, it is necessary to protect the existing cooling landscapes in the suburbs, such as large forest farms, wetlands, and parks, and to treat bara land landscapes is treated with tree-planting or grass-planting to reduce the area of bare land and control the expansion of the impervious surfaces in the built-up area. Reasonable allocation of woodland and water body landscapes, an effective combination of the two layouts, improves the urban heat island effect.

Point 6: All the figures are not clear enough! 

Response 6: Thanks so much for the suggestion. We have redrawn all figures to make it clear enough.

Point 7: It is strongly recommended to proofread the paper thoroughly again.

Response 7: Thanks so much for the suggestion. We have re-proofed the paper.

Kind regards,

The Authors

Reviewer 3 Report

The paper presents a study of landscape pattern and land surface temperature using Landsat remote-sensing data. The percentage of landscape index and patch cohesion index of woodland, water body, and cropland were found to be negatively correlated with LST, while the PLAND and COHESION of impervious surface and barren land were positively correlated with LST. The manuscript is well-written and the analysis is well articulated. I would recommend the publication of this work after addressing my comments given below.

Major comments:

1.     The work is essentially concerned with evolving landscape (land type) and its impact on urban heat islands. Proposing countermeasures is not the major contribution of the work. The title could be refined to reflect more on the major contribution of the work.

2.     The study only involves a city in China. To what extent are you able to generalize the findings reported in this manuscript. An in-depth discussion needs to be added to address this point.

3.     To me, this is mainly a case study concerning the impact of landscape. I’d suggest to add ‘a case study of Chengdu, China’ in both the title and abstract of the manuscript.

4.     Sections 2.2-2.4 mostly present analysis methods reported/used in other studies. I don’t see too much value to document them in such great detail here. I’d suggest to shorten these parts and document some of them in an appendix.

5.     There are quite a lot of recent studies concerning urban heat island and heat mitigations. Just as two examples: (1) Urban Heat Island and Its Interaction with Heatwaves: A Review of Studies on Mesoscale, and (2) Advancement in Urban Climate Modelling at Local Scale: Urban Heat Island Mitigation and Building Cooling Demand. It is worth complementing these most recent studies in the Introduction.

6.     Section 4.3 must be improved to discuss limitations lie in different climate zones, population growth, deployment of mitigation measures in time, etc.

Minor comments:

1.     Fig.1. The caption must be refined to include more information for the three sub-figures.

2.     Fig.2. The caption must be refined to include more information on the landscape, study area, etc.

3.     Fig.3. Please use (very) different colors.

4.     Fig.4. Some texts are not readable. Please use large texts.

5.     Fig.5. Please use (very) different colors for A, B and C.

6.     Fig.6. Please use (very) different colors for 1988, 2000 and 2017.

Author Response

Dear reviewer,

Thank you very much for your valuable time and comments. We were pleased to have an opportunity to revise our paper entitled “Analysis of the Impact of Landscape Patterns on Urban Heat Islands: A Case Study of Chengdu, China”. As instructed, we have carefully considered the reviews and replied to each comment in a point-by-point fashion.

Point 1: The work is essentially concerned with evolving landscape (land type) and its impact on urban heat islands. Proposing countermeasures is not the major contribution of the work. The title could be refined to reflect more on the major contribution of the work.

Response 1: Thanks so much for the suggestion. This study used Landsat data to study the impact of landscape patterns on urban heat islands (UHIs) and temporal-spatial changes characteristics. Therefore, we have revised the title of the article to "Analysis of the Impact of Landscape Patterns on Urban Heat Islands: A Case Study of Chengdu, China" to match the theme with the content better.

Point 2: The study only involves a city in China. To what extent are you able to generalize the findings reported in this manuscript. An in-depth discussion needs to be added to address this point.

Response 2: Thanks so much for the suggestion. In the discussion, we explained that our research results are consistent with those of other urban areas on the regional scale in line 349.

4. Discussion

China promulgated the “National Afforestation Planning Outline for 1989–2000,” promoting greening and afforestation throughout the country. Thus, the area of woodland has increased significantly, showing promising results. Meanwhile, the impervious surfaces district has expanded, occupying much cropland. Since 2000, reduced water body, bare land, and cropland have been transformed into impervious surfaces and woodland. The massive increase in impervious surfaces area has also led to the expansion of the UHIs areas and a more apparent thermal environment. The increase in the woodland area is the best embodiment of the achievements in constructing ecological land (woodland) in the study area.

The main urban area of Chengdu is the core area of Chengdu’s urban agglomeration, with a relatively dense population, rapid economic development, an urbanization rate greater than 70%, and a resident population of approximately 16 million. With the acceleration of urbanization and industrialization, many ecological lands such as woodland and cropland in the study area are used for the construction of infrastructure, and other significant projects, resulting in profound loss of natural resources and apparent expansion of urban heat islands.

The research on the correlation between different types of landscapes and surface temperature shows that forest land and water area have a high negative correlation with temperature. These cold landscapes have a slowing effect on the urban surface heat island. This result is consistent with previous studies on the regional scale [13,53,54]. Therefore, the connectivity of these landscapes and their landscape percentages (PLAND) should be increased. For example, in the woodland landscape of urban boulevards, trees with better canopy cover can be selected to improve the shading effect. However, different trees have different shading capabilities. One crucial criterion is that when considering shading effects, the effect of vegetation on urban airflow must also be considered. Therefore, choosing trees with a higher canopy and better canopy coverage can improve the shading effect and ensure airflow. Further, building connected ditches on both sides of impervious roads or the middle green belt to introduce flowing water can increase urban air humidity and improve the microclimate. The analysis of large and small ditches in the urban area of Chengdu indicated that the water sources in some areas had poor fluidity and appeared to be dry. Treating these areas and introducing flowing water can help alleviate UHIs. The layout of woodland and water body landscapes should be effectively configured and combined, further mitigating the UHI effect. The PLAND of the impervious surfaces is positively correlated with the LST, reflecting that when the impervious surfaces area in the region is larger, there will be a higher temperature. The impervious surfaces contribute more to the urban thermal environment. This result is consistent with the results of previous studies[55,56]. Therefore, the SPLIT of a thermal landscape should be increased. The low value of COHESION is mainly distributed in suburban. These low-value gathering areas are mainly distributed in cold landscapes such as waters and woodlands, and the fragmentation is severe. Therefore, it is necessary to protect the existing cooling landscapes in the suburbs, such as large forest farms, wetlands, and parks, and to treat bara land landscapes is treated with tree-planting or grass-planting to reduce the area of bare land and control the expansion of the impervious surfaces in the built-up area. Reasonable allocation of woodland and water body landscapes, an effective combination of the two layouts, improves the urban heat island effect.

Point 3: To me, this is mainly a case study concerning the impact of landscape. I’d suggest to add ‘a case study of Chengdu, China’ in both the title and abstract of the manuscript.

Response 3: Thanks so much for the suggestion.We've changed the title to better match the theme with the content.

Point 4: Sections 2.2-2.4 mostly present analysis methods reported/used in other studies. I don’t see too much value to document them in such great detail here. I’d suggest to shorten these parts and document some of them in an appendix.

Response 4: Thanks so much for the suggestion. To avoid a lengthy introduction, we rewritten the structure about “2.2. Methods”.Some basic concepts, equations, and techniques for processing LANDSAT data are included in the“Appendix A” file for additional submission.
2.2. Methods

2.2.1. Landscape information visualization methods

According to the ‘Land Use Classification Standards’ and the actual situation of Chengdu's landscape features, we divide Chengdu's landscape into the impervious surface, bare land, cropland, water body, and woodland. The impervious surface mainly includes manufactured impervious surfaces and bare rocks. Cropland mainly includes areas covered by vegetation. Finally, bare land mainly includes cropland without vegetation cover and some unused land. The normalized difference index combined with the decision tree method extracts the landscape pattern information in the remote sensing image. The original image and the high-scoring image in Google Earth are used for post-classification processing and evaluation to improve classification accuracy.

The landscape pattern index can accurately reflect the landscape's structural composition and spatial configuration [43]. Through preliminary research on these indexes provided by FRAGSTATS4.2 software [44], this paper selects the corresponding index from the type and landscape level. In order to show the impact of different types of landscapes on the UHI effect, the splitting index (SPLIT), PLAND, and COHESION at the type level were selected. At the same time, COHESION and SHDI at the landscape level are selected to show spatial variation characteristics of landscape patterns.

2.2.2 Identification of heat islands zone and analysis of space-time change

UHIs are a wide range of climate effects produced by humans changing the urban atmospheric environment. The land surface temperature (LST) is the most direct manifestation of UHIs. Currently, the study of LST mainly uses remote sensing inversion methods, so many data sources are used for inversions, such as Landsat-TM/OLI data [45,46], NOOA-AVHRR data, and ASTER/MODIS [47,48]. According to the difficulty of data acquisition, this paper chooses the NDVI algorithm, which is currently widely used and can be used for the inversion of LST [49]. In addition the estimation process described in NASA’s Landsat5 Scientific Data User Manual and NASA’s Landsat8 Data User Manual is used.

In order to analyze the temporal and spatial changes of the surface heat islands in the study area, the mean-standard deviation method was used to obtain the area range representing different heat levels, and the heat islands area was extracted [50]. Therefore, It is divided into three levels, and the high-temperature zone is defined as the urban heat islands area, and the low-temperature zone is the urban green islands area.

The Heat islands Distribution Index (HIDI) can describe the contribution of different areas in the study area to the thermal environment of the main urban area [24]. The HIDI has a critical value of 1. When the HIDI is greater than 1, the area’s contribution to the thermal environment is more significant than the average contribution; when the HIDI is less than 1, the area’s contribution to the thermal environment is less than the average contribution.

2.2.3. Spatial correlation analysis method

   In order to explore the spatial relationship between the urban landscape pattern and the thermal environment, the Bivariate Moran's I method was adopted at the genre level in 2017, and the Bivariate Local Moran's I method at the landscape level was adopted as an example from 1988 to 2017. The correlation between the landscape index and LST and their spatial aggregation changes are studied. The bivariate Moran's I is statistically significant. This article uses "permutation tests" to evaluate [51,52], which refers to quantifying the observed statistics Relative to the "extreme" degree of its distribution under spatial randomness, the principle is to randomly redistribute the observations of one of the variables, and recalculate statistics for each such random pattern. The 999 "permutation" is used to test its significance. That is, the number of iterations is 999. The larger the value, the more stable the result is, the less likely it is to cause deviation. Set the value of spatial correlation significance to <0.05.

Point 5: There are quite a lot of recent studies concerning urban heat island and heat mitigations. Just as two examples: (1) Urban Heat Island and Its Interaction with Heatwaves: A Review of Studies on Mesoscale, and (2) Advancement in Urban Climate Modelling at Local Scale: Urban Heat Island Mitigation and Building Cooling Demand. It is worth complementing these most recent studies in the Introduction.

Response 5: Thanks so much for the suggestion. We have added some reviews on urban heat islands in lines 46-65 in Introduction and discussed them separately regarding scale, methods, and data selection.

1. Introduction

Today, people are increasingly moving to cities, causing the expansion of urban areas around the world[1]. About 54% of the world’s population lives in cities, and the UN expects that figure to rise to 66% by 2050 [2]. Population growth has increased the intensity of human activity, and urbanization poses increasingly serious problems. Landscape fragmentation caused by human activity (e.g., the extension of roads, railways, and human settlements) has aggravated the expansion of heat islands and affected local climate conditions. In this context, environmental management decision-makers are paying increasing attention to the sustainable management and restoration of urban and suburban green spaces. China’s 14th Five-Year Plan noted that the ecological green wedge and fan-leaf-shaped layout pattern of the ecological park around the city should be used to create a green heart of the city, forming a multi-level urban public center system, shaping the urban form of the beautiful park, and creating a characteristic corridor space to reduce the heat-islands effect.

In 1883, Lake Howard first discovered that the city’s temperature was warmer than that of the surrounding suburbs after comparing the temperature of the city and the suburbs of London [3]. This discovery has attracted widespread attention in academics. In 1958, Manley first proposed the concept of the Urban Heat Island (UHI) [4]. More and more scholars have participated, which has strongly promoted the related research of urban heat islands. At present, the research on UHI is relatively extensive. On the scale, the intensity of surface UHI is estimated based on the global scale and the national scale [5,6]. The environmental sensitivity of surface UHI [7] and the relationship between UHI and outdoor comfort [8] are studied based on local and regional scales. To study the impact of urban green space landscape on seasonal surface temperature based on fine scales such as urban blocks [9] and to study the temporal and spatial changes of the driving factors of UHI [10]. In terms of research methods, the cooling effect of urban vegetation on the UHI is quantified based on the moving crossing method [11], the characteristics of the UHI are analyzed based on the integration method of mobile measurement and GIS spatial interpolation [12], and the impact of water bodies on the UHI is studied based on the weather research and forecasting model method [13]. Simulation of the UHI effect using machine learning methods [14,15]. Because remote sensing data is relatively easy to obtain, most scholars use optical remote sensing images to evaluate surface UHI. However, a few scholars conduct research by measuring the air temperature.

Landscape patterns can change as a result of human activity [16]. The spatial form of the urban landscape determines the urban environment [17], and the increase in the thermal landscape may exacerbate the heat island effect. Common landscape indicators (e.g., patch density, edge density, shape index, aggregation index) have also been used to analyze the effects of green spaces on cooling UHIs [18,19]. Analyzing the shape of the urban landscape through remote-sensing images, combined with landscape measurement, can provide additional accuracy for data representation [20]. The integration of remote sensing and geographic information systems (GIS) is very useful for describing the spatial patterns of urban landscapes. Combining spatial indicators with remote sensing and GIS can facilitate investigating the different structural dimensions of urban landscapes, such as location, distribution, size, shape, and arrangement. These are important variables for quantifying urban expansion [21]. Few of these studies spatialized the landscape index to assess the evolution of the spatiotemporal pattern of the landscape.

Regarding the correlation between urban landscape patterns and UHIs, many studies have used land-cover data to study the effects of different urban landscapes on UHIs [22,23,24,25,26]. Based on Landsat–OLI data, UHIs generated by small and medium-sized cities during different seasons can be quantitatively calculated and their change characteristics analyzed; in this way, the effect of a single land-use type on the UHIs of central cities can be studied [27]. One study used the landscape pattern index to analyze trends in the landscape patterns of UHIs in Suqian [28]. Another study summarized the research history and classification of the UHI effect and specifically reviewed the literature on the effect of landscape patterns on UHIs [29]. Others have studied the relationship from the perspective of green space landscapes or green infrastructure. For example, Nastran et al. analyzed the relationship between the size of European UHIs and the range, shape, and distribution of urban green infrastructure using urban households as a unit [30]. Feyisa et al. studied the cooling effect of parks on the thermal environment in large-scale spaces [31]. The cooling effect of parks on the surrounding environment was positively correlated with normalized difference vegetation index (NDVI) and park area. Others have found that the intensity of UHIs and their correlation with urban green infrastructure are highest in the summer [32,33]. Some studies have found that the intensity of UHIs is most affected by nearby shared green spaces, and considering their spatial distribution is also helpful for analyzing the scope of the influence [34]. Generally, existing studies have shown that agglomerated and continuous large green areas can vigorously promote the reduction of UHIs intensity. In addition, these extensive green areas provide more excellent cooling effects than small green areas [35,36]. There were many studies on the correlation coefficient method in previous academic circles. For example, the Pearson correlation coefficient was used to measure the correlation between surface temperature and landscape indicators [37,38]. However, although many scholars have studied the cooling effect of green space, they are rarely combined with spatial correlation analysis technology.

Urban green infrastructure and impervious surfaces are the main components of urban landscapes, and green spaces play a leading role in improving the human living environment. However, rationally configure green landscapes and impervious surfaces and integrate them with urban planning is a challenging problem. Therefore, this study took the main urban area of Chengdu, China, as the research area and explored the spatial relationship between the main urban landscape pattern and LST using remote-sensing data. It also investigated the expansion of the UHIs and considered whether the spatial layout of the landscape is reasonable. Based on the findings, this study makes policy recommendations for improving UHIs and the rational planning of urban landscape patterns. This can help improve the urban living environment.

Point 6: Section 4.3 must be improved to discuss limitations lie in different climate zones, population growth, deployment of mitigation measures in time, etc.

Response 6: Thanks so much for the suggestion. In the Discussion section, we have made improvements to compare our findings with similar studies and propose deploying mitigation measures in lines 346-375.

4. Discussion

China promulgated the “National Afforestation Planning Outline for 1989–2000,” promoting greening and afforestation throughout the country. Thus, the area of woodland has increased significantly, showing promising results. Meanwhile, the impervious surfaces district has expanded, occupying much cropland. Since 2000, reduced water body, bare land, and cropland have been transformed into impervious surfaces and woodland. The massive increase in impervious surfaces area has also led to the expansion of the UHIs areas and a more apparent thermal environment. The increase in the woodland area is the best embodiment of the achievements in constructing ecological land (woodland) in the study area.

The main urban area of Chengdu is the core area of Chengdu’s urban agglomeration, with a relatively dense population, rapid economic development, an urbanization rate greater than 70%, and a resident population of approximately 16 million. With the acceleration of urbanization and industrialization, many ecological lands such as woodland and cropland in the study area are used for the construction of infrastructure, and other significant projects, resulting in profound loss of natural resources and apparent expansion of urban heat islands.

The research on the correlation between different types of landscapes and surface temperature shows that forest land and water area have a high negative correlation with temperature. These cold landscapes have a slowing effect on the urban surface heat island. This result is consistent with previous studies on the regional scale [13,53,54]. Therefore, the connectivity of these landscapes and their landscape percentages (PLAND) should be increased. For example, in the woodland landscape of urban boulevards, trees with better canopy cover can be selected to improve the shading effect. However, different trees have different shading capabilities. One crucial criterion is that when considering shading effects, the effect of vegetation on urban airflow must also be considered. Therefore, choosing trees with a higher canopy and better canopy coverage can improve the shading effect and ensure airflow. Further, building connected ditches on both sides of impervious roads or the middle green belt to introduce flowing water can increase urban air humidity and improve the microclimate. The analysis of large and small ditches in the urban area of Chengdu indicated that the water sources in some areas had poor fluidity and appeared to be dry. Treating these areas and introducing flowing water can help alleviate UHIs. The layout of woodland and water body landscapes should be effectively configured and combined, further mitigating the UHI effect. The PLAND of the impervious surfaces is positively correlated with the LST, reflecting that when the impervious surfaces area in the region is larger, there will be a higher temperature. The impervious surfaces contribute more to the urban thermal environment. This result is consistent with the results of previous studies[55,56]. Therefore, the SPLIT of a thermal landscape should be increased. The low value of COHESION is mainly distributed in suburban. These low-value gathering areas are mainly distributed in cold landscapes such as waters and woodlands, and the fragmentation is severe. Therefore, it is necessary to protect the existing cooling landscapes in the suburbs, such as large forest farms, wetlands, and parks, and to treat bara land landscapes is treated with tree-planting or grass-planting to reduce the area of bare land and control the expansion of the impervious surfaces in the built-up area. Reasonable allocation of woodland and water body landscapes, an effective combination of the two layouts, improves the urban heat island effect.

Minor comments:

Point 1: Fig.1. The caption must be refined to include more information for the three sub-figures.

Response 1: Thanks so much for the suggestion. We modified Fig.1 to make it more clear.

Point 2: Fig.2. The caption must be refined to include more information on the landscape, study area, etc.

Response 2: Thanks so much for the suggestion. We have refined the title of fig.2.

Point 3: Fig.3. Please use (very) different colors.

Response 3: Thanks so much for the suggestion. We modified Fig.3 to make it more clear.

Point 4: Fig.4. Some texts are not readable. Please use large texts.

Response 4: Thanks so much for the suggestion. We modified Fig.4 to make it more clear.

Point 5: Fig.5. Please use (very) different colors for A, B and C.

Response 5: Thanks so much for the suggestion. We modified Fig.5 to analyze the seven regions' Spatio-temporal changes of heat islands. From 2000 to 2017, the heat island expanded significantly, which was caused by rapid urbanization.

Point 6: Fig.6. Please use (very) different colors for 1988, 2000 and 2017.

Response 6: Thanks so much for the suggestion. We modified Fig.6 to make it more clear.

Kind regards,

The Authors

Round 2

Reviewer 1 Report

The authors have revised the paper in accordance with the reviewers' comments.

Reviewer 2 Report

The manuscript has been improved!